# Numerical Study towards In Vivo Tracking of Micro-/Nanoplastic Based on X-ray Fluorescence Imaging

**DOI:** 10.3390/biomedicines12071500

**Published:** 2024-07-05

**Authors:** Carolin von der Osten-Sacken, Theresa Staufer, Kai Rothkamm, Robert Kuhrwahl, Florian Grüner

**Affiliations:** 1University Medical Center Hamburg-Eppendorf, Department of Radiation Oncology, Medical Faculty, University of Hamburg, Martinistraße 52, 20246 Hamburg, Germany; carolin.baronesse-von-der-osten-sacken@stud.uke.uni-hamburg.de (C.v.d.O.-S.); k.rothkamm@uke.de (K.R.); 2Institute for Experimental Physics, Universität Hamburg, Luruper Chaussee 149, 22761 Hamburg, Germany; theresa.staufer@uni-hamburg.de (T.S.); robert.florian.kuhrwahl@studium.uni-hamburg.de (R.K.); 3Center for Free-Electron Laser Science (CFEL), Luruper Chaussee 149, 22761 Hamburg, Germany

**Keywords:** X-ray fluorescence imaging (XFI), biomedical imaging, micro-/nanoplastic (MNP), palladium, voxel mouse, Monte Carlo simulation, biodistribution, tracking

## Abstract

There is a rising awareness of the toxicity of micro- and nanoplastics (MNPs); however, fundamental precise information on MNP-biodistribution in organisms is currently not available. X-ray fluorescence imaging (XFI) is introduced as a promising imaging modality to elucidate the effective MNP bioavailability and is expected to enable exact measurements on the uptake over the physical barriers of the organism and bioaccumulation in different organs. This is possible because of the ability of XFI to perform quantitative studies with a high spatial resolution and the possibility to conduct longitudinal studies. The focus of this work is a numerical study on the detection limits for a selected XFI-marker, here, palladium, to facilitate the design of future preclinical in vivo studies. Based on Monte Carlo simulations using a 3D voxel mouse model, the palladium detection thresholds in different organs under in vivo conditions in a mouse are estimated. The minimal Pd-mass in the scanning position at a reasonable significance level is determined to be <20 ng/mm^2^ for abdominal organs and <16 μg/mm^2^ for the brain. MNPs labelled with Pd and homogeneously distributed in the organ would be detectable down to a concentration of <1 μg/mL to <2.5 mg/mL in vivo. Long-term studies with a chronic MNP exposure in low concentrations are therefore possible such that XFI measurements could, in the future, contribute to MNP health risk assessment in small animals and humans.

## 1. Introduction

There is a rising awareness of the toxicity of micro- and nanoplastics (MNPs), and yet, fundamental precise information on MNP-uptake and distribution in the organism is missing [1,2,3,4]. For assessing the health risk from exposure to MNPs, which are defined as microplastic particles < 5 mm in diameter and nanoplastic particles < 100–1000 nm in diameter [4,5,6], it is absolutely necessary to obtain exact data on MNP biodistribution [7]. Hence, there is a need for further insights regarding absorption and distribution over the physical barriers of the organism (blood–brain barrier [8], intestinal barrier [9,10,11] and placenta [12,13]), bioaccumulation in the different organs [14,15] and vascular transport [15]. The influence of dose, period of exposure, particle size [1,7], correlation with an altered gut microbiota [16] and co-exposure with other microparticles, pharmaceutics or chemicals [1] on the uptake is highly relevant for MNP uptake studies.

One possible imaging modality for MNP biodistribution studies is X-ray fluorescence imaging (XFI) [17]. While X-rays were initially used for attenuation-based imaging, in XFI, the fluorescence signal for a specific marker that is applied to the target beforehand is detected [17]. Generated for example by a synchrotron, a pencil beam of high-flux, monochromatic X-rays [17] is used to scan the object under investigation. In the following, interaction between incident photons and matter occurs, the most relevant ones being the photoelectric effect and Compton scattering [17,18]. The photoelectric effect describes the absorption of a primary X-ray photon with an energy higher than the absorption edge (often the K-edge) by an electron that will be consequently removed from its shell. When the newly formed “hole” in this shell is refilled with an electron from a higher energy level (outer shell), subsequently, a secondary photon with a characteristic wavelength is emitted [19,20]. Various possibilities for absorption, excitation and refill for the different atom’s shells result in a characteristic spectrum of fluorescence line energies, for example, both the Kα-line and the Kβ-line refer to the refill of the element’s K-shell but from different energy-levels of the higher shells (L, M, …), which then specify the K-line (α, β, …) [19,21]. Based on the quantum mechanical selection rules, Kα1 and Kα2 are the two allowed transitions for the K-shell refill [21]. By filtering the characteristic energy peaks in the detected spectrum, the element of the XFI-marker, as well as its local amount, can be deduced [18]. A limiting factor for the detection of markers is given by the discrimination of the characteristic fluorescence signal from the intrinsic background signal, which is dominated by multiple Compton scattering in XFI [17,18]. The Compton background is minimal between 20 and 30 keV when using an incident beam of around 50 keV [18]. This makes palladium (Pd) a promising marker, as the element’s characteristic Kα fluorescence peak is at 21.12 keV [21]. Also, with the knowledge about the angle dependence of the background arising from Compton scattering, the signal significance can be maximized by adapting the detection angle for imaging [22]. One key advantage of XFI is its high spatial resolution, which is in the range of down to 200 μm for mouse experiments [23], as it is only determined by the selection of the pencil beam’s diameter, whereby a longer scanning time must be accepted with smaller scanning steps [18,23,24]. As shown in immune cell tracking experiments in a mouse before, a 1 mm-pencil X-ray beam with a flux of 8 × 10^10^ incident photons/s and a beam energy of 53 keV provides sufficient fluorescence signal sensitivity to be used for in vivo mouse studies [18]. 

To investigate the MNP biodistribution, detection methods are needed that offer the possibility of in vivo imaging and longitudinal measurements with high sensitivity and high spatial resolution [4,5]. Although there are several methods for MNP detection in the organism, methods such as positron emission tomography (PET) permit in vivo imaging, but the spatial resolution is insufficient (<5 mm) [14,17], and other methods, like light scattering and microscopy [2,25], comply with a high spatial resolution but only in well-prepared tissue samples. Therefore, an imaging method that combines the aforementioned features would be of particular interest. With the focus being to better understand the biodistribution specifically, this work concentrates on simulations mimicking preclinical tracking studies under in vivo conditions (with regard to the applied dose) using a tracer for detection. The feasibility of X-ray fluorescence imaging (XFI) for future in vivo biodistribution studies is presented as a possible promising imaging modality [17,26], which offers the performance of quantitative studies with a high spatial resolution [17,18]. The focus of this paper is a numerical study determining the detection sensitivity limits enabling the design for future preclinical in vivo studies.

In previous studies on the toxicity of MNPs, various effects on the organism have been observed [1,3], while the dependence on dosage, co-exposure and time of exposure is highly relevant [1]. There were reported respiratory disorders [1], gastrointestinal toxicity and correlations with altered gut microbiota [16], increased carcinogenesis [27], especially the risk for lung cancer and colorectal cancer [3], genotoxicity [15] and reproductive dysfunctions [28]. Neurotoxic effects [29] like dizziness, fatigue and headache, but also encephalopathy and unspecified dementia, have been seen, and in vivo studies in rodents have revealed the possible crossing of the blood–brain barrier for MNPs [8]. Additionally, MNP-induced gut barrier dysfunctions, as well as significant correlations of MNP exposure and severity of Inflammatory Bowel Diseases (IBDs), have been described [3,16,30,31]. Since comorbidities like IBD do affect the intestinal barrier, it is interesting to explore the differing risk of MNP exposure for high-risk groups [3,30]. Confirmed microplastics in placenta, both on the maternal and the fetal side, let us think about a probable effect on the newborn [13,32,33]. In relation to cardiovascular toxicity, Marfella et al. revealed a higher cardiovascular risk for patients with MNPs detected in plaque samples from endarterectomy [34], and Persiani et al. described impaired cellular hemostasis, accumulation and toxic effects in the heart [15]. Clearly, several possible toxic effects associated with high MNP exposure are uncovered, but ultimately, it becomes necessary to assess the risk for humans exposed to MNPs in everyday life with relatively low environmental concentrations [4]. When intending to simulate human MNP ingestion in studies, it is important to reproduce environmentally realistic MNP concentrations as well as MNP composition in food or drinking water [1,7]. Human MNP ingestion is estimated to be about 0.1–5 g/week/person [35], and the variety of sources is broad. Microplastics have been detected in 81% of 159 tap water samples globally [36], and even rather unexpected aliments like sugar, salt or honey have shown microplastic contamination [3,37]. In stool samples, Polypropylene (PE) and Polystyrene (PS) were the most detected MNP types [38]. Hence, popular study settings include rodent models that are orally exposed to sphere-shaped PE-/PS-MNPs. It is worth mentioning that Gouin et al. stated the purchase was from five dominating companies and mostly in the shape of spheres [39]. Applied particle sizes mostly varied from about 10 nm to 200 µm, and mice were fed either by one-dose gavage or were chronically exposed to MNP-added food over weeks [1,39]. The study design concerning the selection of MNPs and application can be orientated on this, but several reviews criticized dosages much higher than realistic exposure and those missing variety [1,39]. When planning to perform a tracer-based study, one needs to ensure that firstly, the tracer material itself exhibits the required characteristics for the imaging method, secondly, the possible toxicological profile of the tracer (at least in the applied concentrations) is reasonable, thirdly, the tracer is not changing the MNP’s characteristics that are relevant for their biodistribution-behavior and fourthly, the tracer is linkable to the MNP and not dissolving with the ingestion. 

Expecting to fulfill the named criteria, in this work, simulations are performed with palladium nanoparticles (Pd-NPs). Among other nanoparticles used as tracers in biomedical imaging, e.g., gold nanoparticles (GNPs) [22,24,40] or MoO_2_-particles [26] and iodine as a molecular tracer [41], the element palladium (Z = 46) has shown a promising nanomaterial for tracer applications [21,22,26]. The applicability of such nanoparticles has been explored for in vivo conditions [23,26] and in the context of pharmacokinetic studies [24], (immune)cell tracking [40] or even targeted tumor therapy with labelled CAR (chimeric antigen receptor)-T-cells [22]. In terms of research on the detection limits, differences in penetration depth [17] as well as in significance levels of the corresponding fluorescence peaks at different energies due to different backgrounds in the signal region [18] for Pd compared to other tracer materials are essential. A method for palladium-loading of MNPs has been introduced, with the metal being entrapped inside a polymer [42]. Reasonable biocompatibility for palladium [43] has been reported, but given the knowledge about the toxic potential of many heavy metals [44,45,46] and the fact that palladium belongs to the platinum group elements (PGEs) [44], palladium’s biosafety is not sufficiently established and needs to be proven in further studies [43,44]. 

The aim of this work is to estimate, based on simulations, the detection limits for palladium as an XFI marker under in vivo conditions in a mouse. Supplementarily, for selected setups, the fluorescence counts are calculated manually and compared to the simulation data. The Pd detection limit is determined in 2D projections of harvested voxel organs (colon, liver, kidney, brain) and of the same organs in the voxel mouse model under in vivo conditions regarding the applied radiation dose. The attenuation effect by the surrounding tissue is analyzed, and in correlation, the target position, detection angle and scanning point of the input beam are varied to maximize the fluorescence signal. The estimated detection limits aim to facilitate the planning of the experimental design of subsequent palladium-labelled biodistribution studies and provide a reference for experimental data. 

## 2. Materials and Methods

### 2.1. Geant4 Simulations

The estimation of the expected fluorescence signal for different palladium concentrations is carried out by simulations with the software toolkit Geant4 11.0.3 [47]. A monochromatic polarized X-ray pencil beam of 0.5 mm radius and 10^10^ initial photons with an energy of 53 keV is simulated in various setups including a detector-ring-like setup (see Figure 1) and various targets. The simulated X-ray beam approximates the P21.1 beam line at PETRA III at DESY [48]. To guarantee sufficient signal significance under in vivo conditions, the application of a monochromatic beam is essential, and the incident photon energy is chosen to be relatively far above the palladium K-edge to ensure that only multiple scattered photons would reach the signal energy region. Compton photons need at least 5–6 scattering events to lower the energy close to the signal energy [18]. Scanning the target object in one plane results in 2D projections, with the possibility of locating the marker in a spatial resolution of 1 mm^2^ set by the beam’s diameter, which has been shown to offer sufficient spatial resolution for measurements under in vivo conditions [23,41]. In the plane of linear polarization, the 90° Compton scattering is expected to be suppressed [20].

### 2.2. Detectors

In all setups, a ring of 10 detectors was placed concentrically in the same plane at 6 cm distance to the target. Respecting the distance to the beam at 0°, the detectors were placed in 30° steps (30–150° and 210–330°), as shown in Figure 1. The Geant4-implemented detectors are based on Amptek models SDD7010 (70 mm^2^ FAST SDD^®^; Amptek Inc., Bedford, MA, USA) [49]. The silicon drift diode detector has a 70 mm^2^ detector area that is collimated to 50 mm^2^. With the aim of detecting the spectrum around the characteristic palladium peak for the Kα line at 21 keV [21], this detector fits the Amptek recommended use in experiments with expected detection energies up to 30 keV. 

### 2.3. Setups

#### 2.3.1. Cylinder Mouse Model

To obtain an idea of the palladium concentrations that should be applied in the following setups, in first simulations, a water cylinder (d = 3.4 mm, l = 20 mm) as a simplified model of a mouse colon was placed vertically into a detector ring of 10 detectors. Palladium was added as a homogenous solution in concentrations of (0.18; 0.14; 0.1; 0.01; 0.008; 0.005; 0.001 mg/mL). Referring to the mass of the marker in the beam (1 mm^2^), the detection limit for XFI is preferably given in ng/mm^2^, so 0.18 mg/mL would be equivalent to 180 ng/mm^3^, and summing up the marker’s mass along the transit length at the irradiating point—here, this is d = 3.4 mm—the corresponding concentration would be 612 ng/mm^2^. This setup should serve as a reference for simulations with the harvested colon of the voxel-based mouse model later. The Pd detection limit for the cylinder colon at the 90° (270°) detection angle was 0.001 mg/mL (3.4 ng/mm^2^).

For the exploration of the attenuation effect of the surrounding material and comparison to the simulations with the voxel-based colon, a larger water cylinder (d = 26 mm, l = 60 mm), the “cylinder mouse”, which should represent the coarse geometry of a mouse, and the “cylinder colon” with palladium solution was placed as a sub volume. In this scenario, the effect of the irradiation angle and detection angle should be considered, so simulations were performed with the target placed vertically into the detector ring first, then switched to a horizontal target position in 0°/45°/90° to the X-ray beam. As expected, for these cylinder mouse simulations, the background signal and attenuation effect became more important. The analysis of all 10 detectors of the detector ring for the vertically placed cylinder mouse with a Pd-concentration of 0.01 mg/mL revealed a background angle dependency [22] as demonstrated in Figure 2. Since the fluorescence is emitted isotropically, the signal should not vary for the different detectors if attenuation is equal for all angles, which is the case for the vertically placed cylinder mouse scenario. Consequently, the significance for the orthogonal (90° and 270°) detectors was the highest. Based on these findings, in the following histograms, results for 90° and 270° detectors especially were considered. 

As shown in Table 1, when comparing the results of the cylinder colon model alone to the cylinder colon inside the cylinder mouse for 0.01 mg/mL Pd-concentration, an attenuation effect of 58.7% has been determined, which agrees well with estimations of 56.8% calculated earlier. 

The attenuation effects in dependence of the target placement in the setup were compared by looking at the significance (Z-values). Here, the attenuation-causing material (water) was the same for the whole model, so the focus was on reducing the transit distance between entering the X-ray beam and the emitted fluorescence photons. As expected, due to the linear polarization of the synchrotron beam, the significance Z was the highest for the 90° and 270° detectors with the cylinder mouse being placed vertically in the setup, but for future in vivo measurements with living mice, vertical fixation might be difficult to realize. In the horizontal position of the cylinder mouse, a 45° angle to the beam and detection at 90° would minimize the transit distance and, thus, optimize signal significance, if the examined organ is located centrally in the mouse as is the case in this model. For the presented cylinder mouse setup with 0.01 mg/mL Pd at a 90° (270°) detection angle, significances were Z ≈ 5.7 (6.2) in the horizontal position at 45° to the beam compared to Z ≈ 8.1 (7.3) with the cylinder mouse vertically placed. 

#### 2.3.2. Voxel-Based Mouse Model

The simulated voxel mouse model is based on the 3D whole-body mouse atlas Digimouse (voxel size 0.1 mm, 78.4 × 10^6^ voxels) of a male 28 g mouse by Dodgas et al. [50]. It was implemented in Geant4 with a voxel half size of 0.5 mm and a total of 21 × 100 × 38 voxels in the simulations, as shown in Figure 1. The voxel resolution is higher than the resolution set by the beam diameter of 1 mm and, therefore, does not limit the spatial resolution in the simulations. Each voxel is assigned a class representing an organ, which contains information about its chemical composition and density. The Digimouse data are available at the website of the Biomedical Imaging Group at the University of Southern California [50].

At first, harvested organs were simulated. Therefore, palladium was homogenously distributed into the regarded organ of the voxel mouse, then the surrounding voxels were set to the material air; so actually, just the referred organs of interest were simulated. Simulations in the voxel mouse were planned by measuring the organ dimensions at different irradiating points (±1 mm), see Figure 3. Placing the voxel mouse into the beam was then followed by orientating on these measurements, so that the organ of interest was always in the center of the setup.

The organs that were chosen to be included into the simulations are colon, liver, kidney and brain. This organ selection yields an adequate impression of the attenuation effect, which depends on the depth of the organ in the organism and the surrounding material. Thus, especially the colon was interesting regarding the different irradiating positions from proximal (with a high depth) to distal (more superficial). The selected positions are listed in Table 2. The colon is the most convenient organ to assess the dependence of target depth in the mouse, since the organ cross section at each measuring point is 1.5 mm in the voxel model. The same applies to the angle dependence analysis, with an additional precondition of the colon’s central position in the mouse with proximal measurement. Therefore, these results can be nicely compared to the results from the cylinder mouse model simulations in the question of angle dependence, as shown in Figure 2. In Table 2, the organ cross-sections for liver and kidney (L_O_) in relation to the total mouse diameter (L_M_) at the respective measuring positions are listed. These relations are important for the evaluation of the signal significance in terms of Pd mass in the beam and attenuation-affecting surrounding tissue. The brain simulation was split into measuring medulla, cerebellum and external cerebrum each, as this subdivision was carried out in the mouse voxel model. The Pd concentrations in the brain simulations were set according to the minimum concentrations already known from the other organ simulations, but when no fluorescence counts could be detected, the concentrations were increased to (0.002, 1, 2, 2.5, 5, 10 mg/mL). Since the brain is surrounded by bone, the impact of attenuation is expected to be higher compared to the other organs. The importance of the possible MNP crossing of the blood–brain barrier means that marking the detection limit of MNPs for the brain is of high interest. 

In reference to available data from experiments with a comparable setup, the detection limits for kidney, liver and colon determined in simulations could be compared with experimental measurements, see Table 3 [51]. The mouse experiments were conducted according to Austrian animal welfare legislation (license 2022- 0.257.045), and experimental setups were approved by the local animal ethics committee [51]. In the experimental setup, the harvested organs were measured in an Eppendorf Tube (1.5 mL), so for respecting the possible attenuation of the thin walls, this object (material polyethylene) was added to the simulation. The experimental data deferred from 90°/150°/210° detectors, so the 90°/150°/210° detectors’ simulation results were selected for comparison. The experimental fluorescence counts (N_F_) were scaled with respect to the number of incoming photons, which was 10^10^ photons in the simulations (beam energy in both, simulation and experiment, was 53 keV). For the experiments, limits were given as high and low limits, which correspond to the minimum and maximum signal measured per pixel, depending on the thickness of the organ in the scanning position. These high/low limits were compared to the simulation detection limits, which were defined as significant with Z ≥ 3. As shown in Table 3, the simulation results agree well with measured data.

### 2.4. Simulation Data Analysis

#### 2.4.1. Histogram Analysis

Simulation results deliver a spectrum of photon counts according to their detected energy in keV. In histogram analysis, the characteristic palladium peak at the energy of 21.12 keV [21] is of major interest. As an example, the recorded spectra for the 90°/330° detectors with a Pd concentration of 0.1 mg/mL in the voxel colon setup are shown in Figure 4 where clearly detectable fluorescence peaks at 21.12 keV are observable. 

When searching for the minimal detectable palladium concentration, a significant corresponding fluorescence peak in the predefined signal region is analyzed [40]. Respecting the experimentally determined detector resolution of 150 eV rms, the discrimination of the palladium fluorescence peaks at 21.12 keV (Kα1) and 21.18 keV (Kα2) [21] is not possible. Hence, the signal region of interest is defined around the Kα palladium peak at 21.12 keV [21] with a width of E_Fluo_ ± 3σ, in which the standard deviation σ = 0.15 keV is orientated on the experimentally determined value for the resolution of 150 eV.

The significance in the defined signal region depends on the detected fluorescence counts N_F_ in relation to the number of background counts N_B_ [18,41,52]. Assuming a Poisson statistic for the detected fluorescence counts, with high values expected for N_F_ and N_B_, the Poisson distribution can be approximated with a Gaussian distribution [41], and the signal significance can be described with Z ≈ N_F_/√N_B_ [20,52]. Presuming that N_F_ in the detection limit simulations will not be less than N_B_, a one-tailed hypothesis test is performed [20,53] with the null hypothesis H_0_ being true for the absence of palladium-originated fluorescence photons and the absolute domination of background photons in the observed signal region. A signal is considered as significant when the null hypothesis is rejected with very small *p*-values [53]. Here, for a *p*-value of <0.0015, which corresponds to the commonly used range of Z ≥ 3σ [18,20], the Pd-fluorescence peaks are defined as significant.

In the single organ simulations at the detection angle of 90°/270°, background counts were barely registered, so in the first place, values for N_F_ and N_B_ are not high enough to approximate the statistics to a Gaussian distribution, and aside from that, a significance would not be meaningful to calculate for these scenarios, because without the background, the palladium peak is clearly definable. In that case, thresholds are defined with the lowest palladium concentration added to the organ that resulted in detected fluorescence counts. However, for organ simulations in the voxel mouse, discrimination of the palladium peak from the background becomes necessary.

#### 2.4.2. Estimation of Expected Fluorescence Counts

The number of expected fluorescence counts in the simulation results was calculated for some of the simulated scenarios for verification beforehand, as shown in Table 4. 

The Beer–Lambert Law [54], which describes the attenuation of photons when transiting through a medium, was used for approximating the attenuation effect in the voxel mouse setups. Attenuation depends on the path length × through a tissue for incoming photons (importance of irradiating angle) and outgoing photons (importance of detection angle), with the number of incoming photons N0 and the material specific factors being the mass attenuation coefficient (μ/ρ_t_)_tot_ in cm^2^/g and the material density ρ_t_ in g/cm^3^. NIST provides photon-energy-related mass attenuation coefficients for elements, but also for compounds and mixtures, for example, water [55]. All scenarios were based on irradiation with a monochromatic X-ray pencil beam with a beam energy of 53 keV containing a total number of 10^10^ photons (N0). Taking into account the described dependencies, Formula (1) was used to calculate the effective photon flux N(x) remaining after the attenuation of the incoming photons due to tissue transit [20]. N(x) is the number of photons that is effectively generating the fluorescence counts N_F_ in the following:N(x) = N0 · exp (−(μ/ρ_t_)_tot_ · ρ_t_ · x) (1)

Formula (2) describes the generated fluorescence counts N_F_ [20] estimated based on the absorption cross-section of the fluorescence agent σ_F_ in cm^2^/g, which is material-specific and depends on the incoming photon energy [20,56]. Using the xraylib library for X-ray matter interactions [56], σ_F_ for palladium is 2.69 cm^2^/g for the Kα1 line and 1.42 cm^2^/g for the Kα2 line. The fluorescence counts for Kα (Kα1 + Kα2) are summed up for comparing the results to the simulation counts subsequently; σ_F_ for Kα then is 4.12 cm^2^/g. The transit length d is the diameter of the target including the marker, so the effective fluorescence-generating part is along the beam. In reference to the voxel mouse model, d is the diameter of the simulated organ that includes the homogeneously distributed palladium in the concentration ρ_F_ given in mg/mL. When there is no expected attenuation effect of the surrounding material, like in the single cylinder colon and voxel colon simulations, N_F_ is calculated only with Formula (2): NF = N(x) · σ_F_ · ρ_F_ · d (2)

For the total number of fluorescence events N_F_, again, the attenuation effect must be considered, keeping in mind that the mass attenuation coefficient for the same tissue must be adapted to the energy of 21.12 keV for the Pd-fluorescence photons. Respecting the detector’s relative surface to the total surface of an imaginative sphere at the distance of 6 cm to the target, out of the total outgoing photons, the fluorescence counts for a single detector are then calculated.

When comparing the simulated fluorescence counts to the calculated estimations, deviations for the cylinder colon scenario were <6%; also, for the voxel colon and the cylinder colon in cylinder mouse scenario, the simulated counts accorded with the calculated estimations (<20%). However, for the voxel colon in voxel mouse scenario, deviation increased to ≈60%, but only for the 90° detector, which is correlated to an underestimated attenuation effect of the surrounding tissue. The attenuation effect was estimated by only using the mass attenuation coefficient (μ/ρ_t_)_tot_ and the density ρ_t_ of the NIST “Soft Tissue (ICRU-44)” [55] for surrounding tissue, thus neglecting a possible stronger attenuation by the spinal column in the transit line for the 90° detector. In contrast, the simulations incorporated the different tissue composition and even small sections of bone.

## 3. Results

Based on Geant4 [47] simulations, the detection limits for palladium in the colon, liver, kidney and brain (cerebellum and external cerebrum) were explored, both for harvested organs and in the voxel mouse. Mimicking in vivo measurement conditions for experiments at the P21.1 beamline at PETRA III (DESY, Hamburg, Germany) [48], the simulations were performed with a monochromatic 53 keV X-ray beam with 10^10^ incident photons [18]. Figure 5 presents the resulting detection limits in the different organs in voxel mouse simulations.

In single voxel colon simulations, which would correspond to measurements of the harvested organ in experiments, observed detection limits (defined with a significance Z ≥ 3) were <0.001 mg/mL (<1.5 ng/mm^2^). With the colon placed into the voxel mouse, the detection limit increased to <0.015 mg/mL (<20 ng/mm^2^). The colon in the voxel mouse was simulated in three positions, proximal, central and distal, and revealed relevant significant differences in dependence of the measuring position due to attenuation processes. In simulations with 0.1 mg/mL Pd, the signal significance from distal to proximal was reduced by 68%, mainly influenced by the loss of fluorescence counts, whereas the background signal did not vary relevantly. Hence, the loss of signal significance from distal to proximal is related to the limited penetration depth of the emitted Pd photons [17,22]. 

The detection threshold for the liver was determined to be <0.001 mg/mL (<9 ng/mm^2^) for simulations in the voxel mouse with basal liver irradiation. As expected, because of short surrounding tissue transit distances, when comparing simulations in the voxel mouse to single organ simulations, the attenuation effect in the voxel mouse for liver setups was not as strong as for colon setups. In a central irradiating position of the liver with 0.1 mg/mL Pd for 90° (270°) detectors, the attenuation effect was calculated to −24% (−34%) for the liver setups and −69% (−69%) for the colon setups. 

For kidneys, the detection limit was determined to <0.0015 mg/mL (7.5 ng/mm^2^) in simulations of the right kidney in apical position. When comparing the kidney to the liver and colon simulations with the same palladium concentration of 0.01 mg/mL, which corresponds to 50 ng/mm^2^ in the apical right kidney, 100 ng/mm^2^ in apical liver and 15 ng/mm^2^ in proximal colon, significances (Z) for 90° (270°) detectors were Z ≈ 25 (14) for the kidney and Z ≈ 28 (38) for the liver, while there was no significant signal for the colon. With minimal differences in the background for these simulations, the tracer mass in the X-ray beam mainly influences the signal significance, and having presumed a homogeneous Pd distribution in the organ, that mass depends on the organ cross section in the beam.

Brain simulations revealed detection limits a lot higher compared to the other measured organs. The minimal Pd mass in the beam was in the range of μg compared to ng for the kidney, liver and colon. The detection limit was determined to be <2.5 mg/mL; this concentration corresponds to 16.25 μg Pd in the beam for cerebellum (5 μg of Pd for external cerebrum). The signal significance for Pd concentrations of 10 mg/mL in the brain simulations (cerebellum Z ≈ 13, external cerebrum Z ≈ 17) was in the range of the signal significance that could be reached in the distal colon simulations with only 0.01 mg/mL Pd (Z ≈ 10). In this context, the density of the organ surrounding tissue in the voxel model, in that case, the cranial bone with a density of 1.92 g/cm^3^, becomes relevant. In comparison, the soft tissue density, which is the main surrounding tissue when targeting the liver, kidney and colon, is 1.03 g/cm^3^. Since the photon attenuation depends on the material density (g/cm^3^) and the mass attenuation coefficient (μ/ρ)_tot_ (cm^2^/g), as described in detail in Section 2.3.2, the attenuation effect is expected to be a lot higher for the brain-surrounding tissue, although the transit length is relatively short. 

## 4. Discussion

Having uncovered the Pd detection limits in a voxel mouse model, the opportunities for in vivo XFI biodistribution studies in rodents with palladium tracers, presenting exemplarily the feasibility for exploring MNP uptake, accumulation and excretion, are to be evaluated. 

In this work, observed Pd thresholds in the scanning beam in voxel mouse simulations are <20 ng/mm^2^ for the abdominal organs and <16 μg/mm^2^ for the brain. These detection limits apply for X-ray beam parameters of 1 mm^2^ spatial resolution, 53 keV beam energy and 10^10^ incident photons, thus maintaining a realistic scan time and a radiation dose of ≈300 mGy that is compatible with in vivo mouse experiments [18,41]. Considering that this is the estimated local radiation dose, the effective organ dose in mSv would be remarkably lower [59]. The explored detection thresholds refer to 2D projections of the voxel mouse, so the information about depth is missing here. Though, for evaluation of the biodistribution into the organs over the time, this projection suffices to locate the marker in the corresponding organ. In more detailed examinations, measuring in the third axis enables the mapping of z-coordinates (see Figure 3 for orientation); thus, the identification of the exact accumulation point inside the organ (3D-information) is possible [20]. Yet, projections in two planes implicate higher scanning time and radiation dose delivery [17,26]. 

Using the same X-ray parameters and voxel mouse model as presented in this work, simulations from Kahl et al. with spherical targets (d = 5.5 mm) representing a tumor and with gold (Au) and Pd used as imaging agents, it has already been shown that Pd simulations resulted in higher significances compared to Au-simulations [22]. However, the loss of significance in deeper targets was higher for Pd, which can be explained with the minor penetration depth of palladium [22] and which was also observed in this work when contemplating the significances along the colon in the proximal, central and distal measuring position (loss of significance of −68%). The Au detection thresholds for a subcutaneous target were 0.033–0.1 mg/mL and 0.33–0.1 mg/mL for the target located in the kidney, whereas Pd thresholds were <5 μg/mL for a subcutaneous target and <0.01 mg/mL in the kidney target [22]. Comparing these thresholds to the detection limits of this work, the signal was still significant for Pd concentrations down to <0.0015 mg/mL in the kidney. Another promising XFI contrast agent, molybdenum (Mo), was explored during an 8-week follow-up tracking molybdenum oxide nanoparticles (MoO_2_)—but unrelated to nanoplastic experiments—in a longitudinal in vivo mouse study [26]. The opportunity of longitudinal measurements with XFI is emphasized at this point, because other in vivo imaging modalities have limitations, like PET being limited by the half-life in the range of hours of applied radionuclides [25,26]. Linked to XFI markers, MNPs can be tracked in vivo over weeks as long as they remain in an organism, so the analysis of long-term accumulation in organs and chronic exposure experiments are possible. The detection limit for MoO2 nanoparticles was estimated to be 0.05 mg/mL when scanning the mouse with a resolution of 200–400 μm with a 24 keV semi-monochromatic pencil-beam [26] where the photon flux at focus was measured to be 2.8 × 10^7^ photons [26]. The average XFI radiation dose for a 15 min 2D-full body scan (200 μm × 200 μm pixel size) in this setting was estimated to be 1 mGy [26]. While longer scanning times per focus point indeed increase the signal significance, radiation dose can be reduced with shorter scanning times [17], and therefore, additionally, for practical applications the benefit of a shorter measuring time would be the crucial point. Finally, the mouse well-being, which is a fundamental requirement for longitudinal measurements, was explored in the study to assess the impact of XFI-measurements and confirmed the feasibility at least for preclinical in vivo mouse studies [26]. 

Liu et al. [43] critically explored the perspectives for palladium use in biomedicine, also questioning the biosafety concerns coming along with Pd applications. In the review, the broad use of Pd in imaging (SPECT, PET, CT and MR) and, moreover, as therapeutic agent becomes clear. Particularly with regard to the combination of imaging and therapy, for example, in the form of targeted drug delivery, the relevance of XFI as a precise imaging method due to its high resolution is to be highlighted here. One therapeutic approach for cancer therapy using Pd is photothermal therapy (PTT), which showed synergistic effects combined with chemotherapy, radiotherapy and immunotherapy [43]. The palladium tracer enables controlled therapy and additionally enhances specific uptake and accumulation in the target [22,42]. Assessing the Pd toxicity profile, cytotoxic and pro-inflammatory effects for in vitro studies are discussed in a review from Leso et al. [45]. In vivo mouse experiments with oral feeding or intraperitoneal injection of Pd nanosheets (5–80 nm) showed no remarkable lesions in H&E stained samples and a renal excretion for particles < 30 nm has been observed; however, for particles > 30 nm, a significant Pd-accumulation in liver and spleen was proven [60]. Future studies using Pd as an XFI agent should consider that data from in vivo experiments for low-dose and coated Pd [43] application, which would be incorporated conditions for XFI experiments [42], are insufficient [45] and should be investigated.

Comparison with alternative XFI agents and XFI setups shows that palladium offers a promising low detection limit, yet attenuation issues need to be considered more for Pd than for Au (when contemplating K-shell fluorescence yields) [22]. In this work, simulations fully incorporated the attenuation effects. Photon flux, scanning time and incident photon energy are key points to adapt the X-ray beam characteristics for maximization of signal significance and minimization of radiation dose. XFI therefore enables to perform long-term studies on MNP distribution in accordance with mouse well-being in preclinical studies. 

By loading MNP-particles with the XFI-marker, the connection of XFI and MNP-tracking is warranted. Mitrano et al. doped nanoplastic particles (<190–210 nm) [42,51] with chemically entrapped palladium to improve MNP detection in complex media [42]. The even Pd distribution over the particles was proven, and Pd leaking over time was assessed. The maximum concentration of leached Pd after eight weeks was 0.05 μg/L [42]; therefore, under corresponding conditions, applicability for MNP toxicological risk assessment is given, and palladium’s toxicity impact can be neglected. Bonding to polyacrylonitrile (PAN), palladium was forming the core of spheres with a polystyrene (PS) shell in different shapes, which was controlled by scanning electron microscopy (SEM) [42]. Moreover, particle surface charges (Zeta potential), influencing the MNP’s biodistribution [2,14,61], were measured for palladium-doped particles and showed negative charges of around −50 mV [42], which is in the range of charges for pure polystyrene beads at neutral or high pH [61]. When using materials produced from Mitrano et al., detected 1 μg/L Pd would correspond to 1.24 × 10^11^ PS-shell particles/L or approximately 1 μg nanoplastic/L [42]. Hence, at a concentration of the detection limit determined here for Pd in the voxel liver of <1 μg/mL, 1.24 × 10^11^ particles/mL (<1 μg/mL) of palladium-loaded nanoplastics (Pd-MNPs) would be detectable. 

Referring to the results of previous biodistribution studies, the expected MNP distribution and detected amounts in experiments are discussed, also comparing the applied methods in those studies to XFI-based studies. An overview of the in vivo tracking imaging modalities presented can be found in Table 5. While PET and BLI have been already applied in MNP biodistribution studies, there are also new approaches for in vivo tracking studies like magnetic particle imaging (MPI) as an in vivo imaging modality [62]. For example, Wijesinghe et al. introduced MM-ODT (magneto-motive optical Doppler tomography) using SPIO (supramagnetic iron-oxide) nanoparticles as tracers for tracking melanoma cells [63]. The use for MNP tracking could be possible and should be further investigated.

An in vivo mouse PET study from Im et al., with orally applied ^64^Cu-labelled polystyrene (0.2–0.3 μm, 2.5 mg), revealed transit to all tested organs (stomach, intestines, spleen, liver, heart, blood, lung, kidney, bladder, testis, brain) and accumulation in liver and testis after 48 h, with measurements at 5 time points in 48 h [14]. Parameters for biodistribution were %ID/g (percentage of injected dose), SUV_max_ (Standard uptake value) and AUC (Area under the curve) [7], which would also be useful parameters for MNP XFI-studies. Another PET study from Keinänen et al. applied a rather low dose of 5 mg/kg ^89^Zr-labelled polystyrene (20 nm–6 μm), measured at 4 time points in 48 h, and observed that NPs predominantly remained in the gastrointestinal tract and were finally excreted [64]. When comparing PET studies to XFI studies, they are limited by a low spatial resolution of <5 mm in vivo [17], and the applicability for longitudinal studies that would surpass the half-life of the used radionuclides is impossible. Regarding the presented studies, it would be interesting if organ accumulation would be present at later time points > 48 h. Also, insufficient spatial resolution impedes exact MNP location.

Another approach for in vivo tracking of MNPs is presented by Liang et al., using in vivo optical fluorescence imaging, which is mainly limited by the penetration depth of the linked fluorescence marker [1]—in clear contrast to X-ray fluorescence imaging, where attenuation is much lower. After a single dose gavage (250–500 mg/kg body weight), accumulation of labelled polystyrene MNPs could not be detected in vivo, although organ analysis after 24 h ex vivo confirmed MNPs in inner organs [10]. The total bioavailability here was estimated to be 0.46–6.16%, depending on the particle size (50–5000 nm). This study also conducted an MNP-toxicity assessment with a 28-day repeated exposure (2.5–500 mg/kg body weight) and revealed relevant toxic effects on the organism, including an intestinal barrier dysfunction [10]. Considering the estimated human MNP ingestion of 0.1–5 g/week/person [35] by Senathirajah et al., the ingested MNP mass per day would be >14 mg/d, so even the 2.5 mg/kg body weight dosage applied daily in a 28-day exposure [10] is notably higher than under realistic environmental exposure conditions and should be respected for planning an MNP XFI study.

Loading MNPs with Pd in nm to μm range has been shown to be possible [42]. The three presented in vivo biodistribution studies did not give any information about detection limits, neither about exact accumulated mass of MNPs in the organs. Yet, with regard to the estimated bioavailability and oral dosage from Liang et al., bioavailable MNP concentrations would be in the range of 1.15 mg/kg to 30.8 mg/kg body weight. With reference to the Pd-MNP detection limit of <1 mg/L, these concentrations can be detected with XFI.

Alternative MNP detection methods are tracer-free and sample-analyzation-based studies including various forms of microscopy, light scattering, spectroscopy or mass spectrometry [2,25,65]. High sensitivity and specificity as well as applicability for MNP studies have been shown for these methods over the years [2,25,65]. Not being influenced by an added tracer material, the biodistribution of MNPs is realistic compared to the environmental exposure. Direct MNP proof in human samples, for example, in blood [34,66], placenta [13,67], liver [68], feces [38] and urine [69] provides information about uptake, distribution and excretion. Internal exposure can be related to external exposure, assessing the environmental concentrations individually [70]. Particles down to sizes of <3.3 μm could be detected in human liver tissue [68] and mass spectroscopy detection thresholds are in the range of μg [65]. For example, in contrast to the described in vivo tracking studies, Kopatz et al. showed the transit of orally applied polystyrene nanoparticles (5 nm)—but no micrometer sized particles (0.29 μm to 9.55 μm)—over the blood–brain barrier in a short-term uptake study in mice by harvesting the brains 2 h or 4 h post gavage and detecting the MNPs with fluorescent microscopy analysis in tissue samples [8]. Considering the importance of the different uptake and accumulation properties depending on the particle size, tracking particles in the nanometer range becomes even more interesting. Yet, when considering tracer-based methods again, the difficulty of incorporating the tracer the smaller it is must be taken into account. 

**Table 5 biomedicines-12-01500-t005:** With the focus on MNP biodistribution research, X-ray fluorescence imaging (XFI) is compared to the two other introduced imaging modalities, positron emission tomography (PET) and bioluminescence imaging (BLI).

	XFI	PET	BLI
Spatial Resolution	in vivo < 1 mm [17]	<5 mm [14,17]	>mm [10]
Labelling	Broad variety of elements as tracer (X-ray fluorescence is a property of every element [20]), e.g., Au [22], Pd, MoO2 [26], I [41]	Need for biocompatible radionuclides with a relatively long half-life, e.g., ^64^Cu [14], ^89^Zr [64]	Labelled MNPs (polystyrene spheres) are available on the market for purchase [10]
Accessibility of the Imaging Technology	Synchrotron availability limited [17], alternatives: liquid-metal-jet X-ray source [71], X-ray source combined with mosaic crystal optics [72], laser wakefield accelerators [73]	PET/CT scanners for small animal studies are available for preclinical applications [14,64]	Multispectral imaging systems for in vivo studies in mice are available [10]
Limitations for longitudinal imaging in vivo	Background signal (Compton) [17,20], the scanning beam size determines measuring time and spatial resolution [17], radiation dose depends on the scanning time, beam energy and photon flux [17]	Radionuclide decay determines max. measuring time, temporal and spatial resolution [17,25]	Autofluorescence (Background signal), low penetration depth and low sensitivity [1,20,64], half-life of fluorescent label [25]

XFI features that have been presumed for measuring under in vivo conditions in this work are provided by the synchrotron-produced high flux monochromatic X-ray beam at PETRA III (DESY, Hamburg, Germany). Synchrotron facilities are huge, expensive and have a very limited access; nevertheless, research on compact X-ray sources hold promise for wider availability and applicability, including preclinical use for XFI [17]. Shaker et al. performed an in vivo mouse study with a 24 keV semi-monochromatic pencil-beam generated by a liquid-metal-jet X-ray source [71] in combination with a multi-layer mirror to focus the beam [26]. Another benchtop XFI system has been introduced by Kumar et al., who also concentrated on focusing the X-ray beam, but with mosaic crystal optics. The X-ray source here is a microfocus bremsstrahlung tube [72]. Alternatively, laser wakefield accelerators (LWFAs) as compact X-ray sources are based on operation with high-power lasers, and by making use of the principle of inverse Compton scattering, these sources provide high-intensity quasi-monochromatic beams with tunable energy in the hard X-ray range, depending on the electron energy [17,73]. With the clear aim of expanding the availability of XFI sources, cone beam irradiating systems using diagnostic X-ray tubes allow experiments with a relatively short scan time and employing a lead collimator and additional filters result in a hardened beam, dose reduction and quasi-monochromatization [17,74].

From the medical point of view, broad research on MNPs over the last years finally aims to achieve a comprehensive risk assessment and, more precisely, converges into the determination of orienting benchmark doses in humans [7]. Presented from Noventa et al. a risk assessment framework suggests to estimate the human risk in the form of modelling approaches which integrate empirical and MNP analytical data [70]. Experimental quantitative data on biodistribution from XFI studies in rodents would contribute to required input data [7,70]. In a review from Wardani et al., a literature search for studies focusing on MNP biodistribution in specific, 23 experimental MNP biodistribution studies (12 in vivo studies in rodents) accorded with the inclusion criteria for such a risk assessment model: the proposed physiologically based pharmacokinetic (PBK) model, which unites absorption, distribution and excretion data [7]. The studies presented, from Im et al., Keinänen et al. and Liang et al., are among those included in vivo studies [7]. Inclusion criteria define the quality of the study (particle characterization and experimental study design) but also the applicability for biodistribution modeling, in which biodistribution values and metrics are more important than reported toxicity effects [7]. Comparable and standardized biodistribution parameters like AUC and %ID/g [7,14] become important here. This quantitative data can be acquired with XFI measurements and, in contrast to other in vivo imaging modalities, with the advantage of a more precise localization due to the high spatial resolution. Moreover, XFI makes long-term studies with a chronic MNP exposure in low concentrations possible. Therefore, indeed, XFI, for example with Pd as a tracing agent, is a promising imaging modality for assessing the MNP biodistribution in mice and might, in the future, contribute to the health risk assessment in humans.

## 5. Conclusions

XFI has the potential to become a novel method for preclinical MNP biodistribution measurements, allowing quantitative, non-invasive, and spatially high-resolution long-term studies in small animals with a chronic MNP exposure at low concentrations. Information on uptake and distribution with a low-dose exposure would contribute to the human health risk assessment when orientating on environmentally realistic MNP concentrations in experiments. For the introduced XFI tracing agent palladium, the numerical studies have shown promising low thresholds down to a few ng/mm^2^ in voxel mouse simulations. Future experimental XFI studies will show whether Pd-labelled MNPs can be detected in vivo with the high sensitivity predicted by the simulations. In vivo measurements should also further investigate the effects of the potential toxicity of Pd as well as Pd leakage.

## Figures and Tables

**Figure 1 biomedicines-12-01500-f001:**
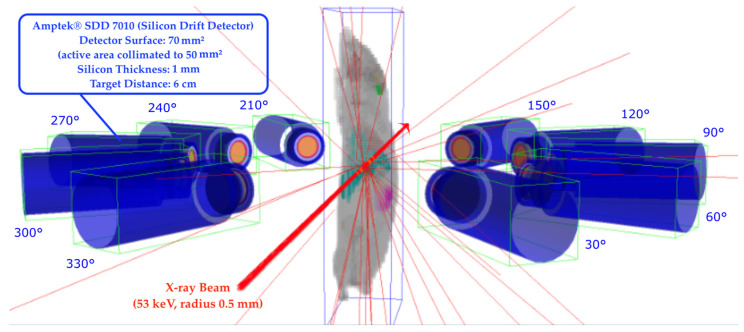
Visualization of the complete setup in Geant4 for planification of simulations using the voxel mouse model as target. The Amptek detectors are placed in 30° steps with respect to the X-ray beam 0°. The mouse position is adapted so that the liver is located directly in the beam.

**Figure 2 biomedicines-12-01500-f002:**
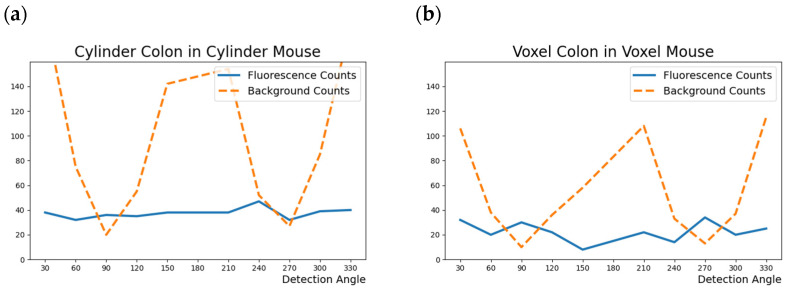
Angle dependence for fluorescence counts (N_F_) and background counts (N_B_) with 0.01 mg/mL palladium distributed homogenously in the colon in (**a**) cylinder mouse model and (**b**) voxel mouse.

**Figure 3 biomedicines-12-01500-f003:**
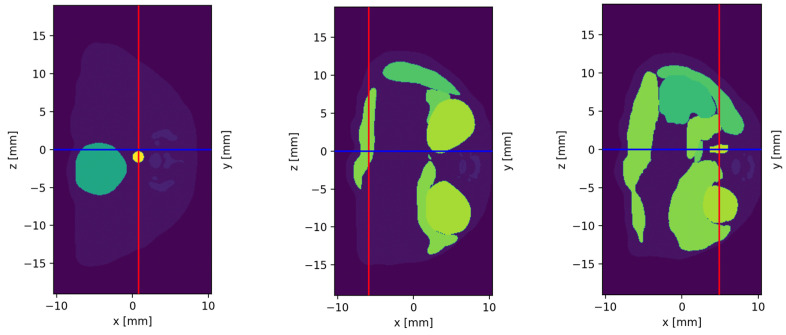
Planification of simulations in the voxel mouse. At each scanning point (crossing of the red and blue line) the cross-section (L_O_) of the target organ (from left to right: colon, liver, kidney) and the total cross-section of the mouse (L_M_) in the beam is measured in mm (±1 mm) along the z-axis.

**Figure 4 biomedicines-12-01500-f004:**
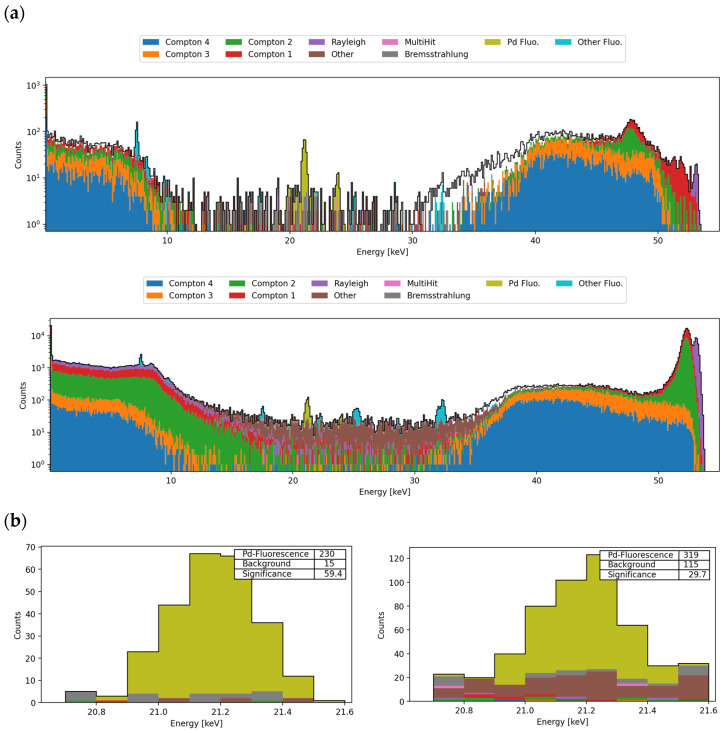
(**a**) Recorded spectra for simulations of the voxel colon (0.1 mg/mL Pd) in the voxel at the detection angles 90° (**up**) and 330° (**down**). The beam (53 keV, 10^10^) is coming from ventral and the colon is scanned in distal position. The corresponding Pd fluorescence peaks at 21.12 keV are shown in green. (**b**) The Kα signal region of 21.12 keV ± 3σ (σ = 0.15 keV) with the corresponding Pd fluorescence peaks is shown for the detection angles 90° (**left**) and the 330° (**right**).

**Figure 5 biomedicines-12-01500-f005:**
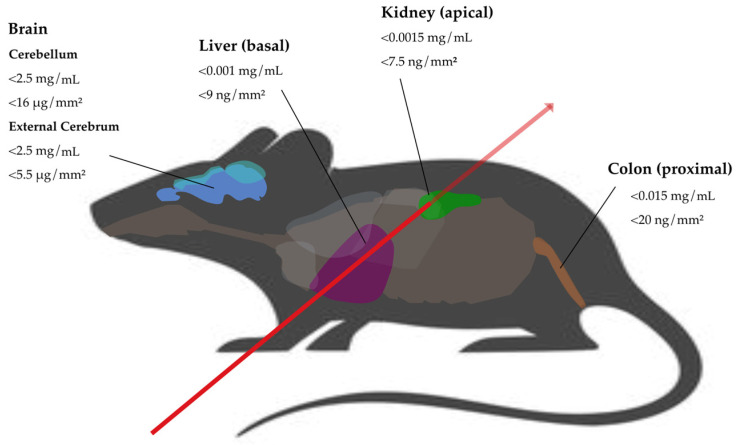
Detection limits for voxel mouse organs (brain, liver, kidney and colon) given for palladium concentrations in the organ in mg/mL and palladium mass in the beam in ng/mm^2^ (μg/mm^2^). Data were acquired in simulations with the beam coming from lateral as indicated by the red arrow using 53 keV and 10^10^ photons. Taken and modified from [57,58].

**Table 1 biomedicines-12-01500-t001:** The estimated fluorescence counts (Estimated N_F_) and the fluorescence counts from simulations (Simulated N_F_) are compared for the cylinder colon model and the voxel colon model. Pd in a concentration of 0.01 mg/mL is homogeneously distributed in the colon models, simulated either harvested or inside the mouse model. The X-ray beam is set to 10^10^ photons, 53 keV, and is coming from lateral. The detected fluorescence counts for the 270° detector are given.

	Estimated N_F_	Simulated N_F_	Deviation from Estimation
Cylinder Colon	155	146	−6%
Cylinder Colon in Cylinder Mouse	67	64	−4%
Attenuation effect	−57%	−62%	
Voxel Colon	68	62	−9%
Voxel Colon in Voxel Mouse (proximal)	22	18	−18%
Attenuation effect	−68%	−71%	

**Table 2 biomedicines-12-01500-t002:** With the beam coming from lateral, the cross-sections (L_O_, L_M_) as shown in Figure 3 were measured for the organs of interest. Also, with the beam coming from ventral, the colon simulations were planned in proximal, central and distal position.

Organ	x	y	z	L_O_ (Organ) in mm in Beam-Line	L_M_ (Mouse) in mm in Beam-Line
Colon (proximal)	−0.75	20	0	1.5	25
Colon * (proximal, central, distal)	0	20	0	1.5	16
	0	30	0	1.5	14
	0	39	0	1.5	9
Liver (basal)	6	9	0	9	26
Kidney (right, apical)	−5	6	0	5	22

* Colon measured from ventral.

**Table 3 biomedicines-12-01500-t003:** The experimentally determined thresholds in ng/mm^2^ at the angles of 90°/150°/210° [51] and the detection limits from Geant4 simulations are compared. The setup consisted of harvested organs (kidney, liver and colon) measured in an Eppendorf Tube (1.5 mL). In the simulation, the beam was set to 10^10^ photons of 53 keV incident energy, whereas the detected fluorescence counts from experiments were scaled to 10^10^ incident photons for comparison.

Organ	Detector	Experimental Data: Scaled Limits Low/High (Average) in ng/mm^2^	Simulation Limits in ng/mm^2^	Deviation from Experimental Data in ng
Kidney	90° 150° 210°	1.7/2.6 (2.2) 3.9/4.4 (4.2) 6.5/9.7 (8.1)	<1.5 <5 <5	<0.7 <0.8 <3.1
Liver	90° 150° 210°	1.6/2.4 (2) 3.5/5 (4.2) 6.6/9 (7.8)	<7 <17.5 <17.5	<5 <13.3 <9.7
Colon	90° 150° 210°	1.7/2.7 (2.2) 2.8/4.9 (3.9) 5.3/9.6 (7.4)	<3 <4.5 <4.5	<0.8 <0.6 <2.9

**Table 4 biomedicines-12-01500-t004:** In dependence of the homogeneously distributed tracer Pd (concentration given in mg/mL), the estimated fluorescence counts (Estimated N_F_) and the fluorescence counts from simulations (Simulated N_F_), with the X-ray beam set to 10^10^ photons, 53 keV, are listed. Simulated N_F_ are the detected counts from the 90°/270°detector, and for both detectors, the deviation of Simulated N_F_ from Estimated N_F_ is shown.

	Pd Concentration in mg/mL	Estimated N_F_	Simulated N_F_ 90° (270°)	Deviation from Estimation
Cylinder Colon	0.18 0.14 0.01	2786 2167 155	2691 (2684) 2099 (2091) 154 (146)	−3.4% (−3.7%) −3.1% (−3.5%) −0.6% (−5.8%)
Voxel Colon	0.1 0.01 0.005	683 68 34	740 (702) 80 (62) 29 (27)	+8.3% (+2.8%) +17.6% (−8.8%) −14.7% (−20.6%)
Cylinder Colon in Cylinder Mouse	0.01	67	65 (53)	−3% (−20.9%)
Voxel Colon in Voxel Mouse *	10 1 0.01	22,113 2211 22	8782 (17,068) 927 (1780) 8 (18)	−60.3% (−22.8%) −58.1% (−19.5%) −63.6% (−18.2%)

* Beam is coming from lateral.

## Data Availability

The data presented in this study are available on reasonable request.

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
