# Peer review of "Numerical Study towards In Vivo Tracking of Micro-/Nanoplastic Based on X-ray Fluorescence Imaging"

_biomedicines, 2024, doi:10.3390/biomedicines12071500_

Round 1

Reviewer 1 Report

Comments and Suggestions for Authors

Review for the manuscript:

Entitled: " Numerical Study towards in vivo Tracking of Micro- /Nanoplastic based on X-ray Fluorescence Imaging"

for Biomedicines.

With ID: biomedicines-3072991

General comments

Comments for the Authors,

This work is well within the scope of Biomedicines and it may be of interest to a specific audience, with good references to follow.  

The topic is very interesting since micro and nano plastics pollution is of paramount importance in our days. The article is written well however in a bizarre order with the materials and methods sections after the results. Most of the comments and questions that arose to me by reading the text were answered in the following paragraphs. The reason for this may be the reversed order of the paragraphs. There is a one-page results section in page 4, then the discussion and all the actual results are in the materials and methods section. Is this modification approved by all authors and why? Furthermore, more effort should be given for the justification of the findings of this work, especially since the current work refers to 2D and not to 3D projections. iThenticate report is very low.

Due to this, I have opted to recommend a Major Revision for the current version of the manuscript.

Specific comments

Introduction

P2: ‘and spatial and temporal resolution and characterization of MNPs are needed’ Please revise.

Materials and Methods

P9: 4.1. Geant4 Simulations

In this paragraph more, details can be provided. For example: ‘is simulated in various setups including a detector ring-like setup and various targets’ Authors could provide a detailed figure (or revise figure 3 with more details) showing the detector with its components along with relative details.

Furthermore ‘to assure that only very few multiple scattered photons would reach the signal energy region’ This is a very general statement. Please provide more accurate values.

P11: ‘Here, the stimulations’ Please correct.

Author Response

Reaction to Review 1 for the manuscript: 

Entitled: " Numerical Study towards in vivo Tracking of Micro- /Nanoplastic based on X-ray Fluorescence Imaging"

for Biomedicines.

With ID: biomedicines-3072991

Dear Editor, dear reviewers,

We are really thankful for your very valuable feedback which we have taken step by step for further improving our manuscript. Please find below our response to all of your points (copied from the reports and given in italics). Changes in the manuscript are highlighted in blue.

Comments for the Authors

This work is well within the scope of Biomedicines and it may be of interest to a specific audience, with good references to follow.  

The topic is very interesting since micro and nano plastics pollution is of paramount importance in our days. The article is written well however in a bizarre order with the materials and methods sections after the results. Most of the comments and questions that arose to me by reading the text were answered in the following paragraphs. The reason for this may be the reversed order of the paragraphs. There is a one-page results section in page 4, then the discussion and all the actual results are in the materials and methods section. Is this modification approved by all authors and why? Furthermore, more effort should be given for the justification of the findings of this work, especially since the current work refers to 2D and not to 3D projections. iThenticate report is very low. 

Due to this, I have opted to recommend a Major Revision for the current version of the manuscript.

As we had previously submitted our manuscript to IJMS, we followed the given structure of this journal. Since the manuscript was forwarded to Biomedicines during the submission process, we did not revise the structure until then. It is perfectly understandable that the reading is more comprehensible when the methods are explained before presenting the results, so we have adapted the structure of the manuscript to 2. Methods, followed by 3. Results, 4. Discussion and 5. Conclusion.

In P5-8 of the discussion section we explained that 2D-projections will be sufficient for the location determination of MNPs in the organs, but that it is also possible to measure in a third axis to complement the information of depth (3D-information).

We have thus added the following into our manuscript.

“In more detailed examinations, measuring in the third axis enables the mapping of z-coordinates (see Figure 3 for orientation), thus, the identification of the exact accumulation point inside the organ (3D-information) is possible [20]. Yet, projections in two planes implicate higher scanning time and radiation dose delivery [17,26].”

Specific comments

 Introduction

P2: ‘and spatial and temporal resolution and characterization of MNPs are needed’ Please revise.

We revised and adapted the phrase to:

“To investigate the MNP-biodistribution, detection methods are needed that offer the possibility of in vivo imaging and longitudinal measurements with high sensitivity and high spatial resolution [4,5].“

Materials and Methods

P9: 4.1. Geant4 Simulations

In this paragraph more, details can be provided. For example: ‘is simulated in various setups including a detector ring-like setup and various targets’ Authors could provide a detailed figure (or revise figure 3 with more details) showing the detector with its components along with relative details.

We have complemented the figure (now Figure 1) showing the Geant4-setup. For that, we have added details of the detector features and the irradiating beam parameters.

Furthermore ‘to assure that only very few multiple scattered photons would reach the signal energy region’ This is a very general statement. Please provide more accurate values. 

We specified this statement with the complementary information that there is a simultaneous loss of energy for multiple scattered Compton photons.

“…and the incident photon energy is chosen to be relatively far above the palladium K-edge to ensure that only multiple scattered photons would reach the signal energy region. Compton photons need at least 5-6 scattering events to lower the energy close to the signal energy [18].“

P11: ‘Here, the stimulations’ Please correct.

We changed this section to the following phrases:

“The brain simulation was split into measuring medulla, cerebellum and external cerebrum each, as this subdivision is done in the mouse voxel model. The Pd concentrations in the brain simulations were set according to the minimum concentrations already known from the other organ simulations, but when no fluorescence counts could be detected, the concentrations were increased..”

Reviewer 2 Report

Comments and Suggestions for Authors

Carolin Von and the team have demonstrated the capability of in vivo Tracking of Micro- /Nanoplastic based on X-ray Fluorescence Imaging. The title is kind of a timely concern as nanomaterials, such as nanoplastics and other relative nano particles. However, the  current content of the manuscript needs extensive revisions to provide a robust reliability of the results. I have given few remarks to enhance the quality.

Major Remarks

1. The study says in vivo tracking, which does not provide any evidene of in vivo testings. Can authors explain this?

2. Can the resolution of X-ray or fluroescence detect the nanoparticles? I doubt about the capability. So before claining about spatial resolution, 

I suggest authors to be well aware about the spatial resolution of the system.

3. Also, when it comes to the detection of medical imaging devices with a higher spatial resolution for microscopic and cross-sections,

 Optical Dopplere imaging and photoacoustic imaging

plays a vital role. I suggest to discuss the differences and how capable of the proposed methods over aforestated methods. 

It is worthy to cite and compare below references.

 a. In vivo imaging of melanoma-implanted magnetic nanoparticles using contrast-enhanced magneto-motive optical Doppler tomography

 b. Fully waterproof two-axis galvanometer scanner for enhanced wide-field optical-resolution photoacoustic microscopy

4. The novelty of the method is not properly elaborated.

6. rather than other original articles, this paper is more closer to a review paper. The simulation results can be incorporated with real data.

7.How successful this method is going to be when it is applied to the real scenario? Can author discuss about the real-feasibility as it is hard to understand  with simulation data.

Comments on the Quality of English Language

Moderate Checking is required.

Author Response

Reaction to Review 2 for the manuscript: 

Entitled: "Numerical Study towards in vivo Tracking of Micro- /Nanoplastic based on X-ray Fluorescence Imaging"

for Biomedicines.

With ID: biomedicines-3072991

Dear Editor, dear reviewers,

We are really thankful for your very valuable feedback which we have taken step by step for further improving our manuscript. Please find below our response to all of your points (copied from the reports and given in italics). Changes in the manuscript are highlighted in blue.

Major Remarks

1. The study says in vivo tracking, which does not provide any evidene of in vivo testings. Can authors explain this?

Our work does not provide any experimental data from in vivo studies; however, we present results of numerical studies which are the basis for any XFI-based nanoparticle tracking studies. We have evaluated these findings in the context of future in vivo tracking studies in mice, considering the selection of the X-ray beam parameters for the experimental setup like for example sufficient signal sensitivity with a reasonable radiation dose relevant for in-vivo studies, which is the reason for why we use the term “in-vivo”.

2. Can the resolution of X-ray or fluroescence detect the nanoparticles? I doubt about the capability. So before claining about spatial resolution, I suggest authors to be well aware about the spatial resolution of the system.

We are well aware about the high spatial resolution of XFI that is emphasized in our manuscript. As explained in the manuscript, the spatial resolution is defined by the X-ray beam diameter, which was set to 1 mm in our simulations. For palladium as tracer material, with the X-ray beam parameters that we have selected in our simulations, the penetration depth and signal sensitivity of Pd fluorescence has shown to be clearly high enough for detecting accumulated MNPs in very low concentrations down to μg/mL. We have also referred to own experimental work in which the spatial resolution for scanning of murine thyroids under in-vivo conditions has been shown (Körnig et al., Sci Rep 2022).

3. Also, when it comes to the detection of medical imaging devices with a higher spatial resolution for microscopic and cross-sections, Optical Dopplere imaging and photoacoustic imaging plays a vital role. I suggest to discuss the differences and how capable of the proposed methods over aforestated methods. It is worthy to cite and compare below references.

(a) In vivo imaging of melanoma-implanted magnetic nanoparticles using contrast-enhanced magneto-motive optical Doppler tomography

(b) Fully waterproof two-axis galvanometer scanner for enhanced wide-field optical-resolution photoacoustic microscopy

We have revised the proposed references and cited the study of Wijesinghe et al. (a), because magnetic particle imaging (MPI) represents an in vivo tracking imaging modality that is comparable to XFI, PET and BLI that we have contemplated more closely in our work. Regarding the reference b, we decided to not include this imaging method, because it is a label-free scanning method, so the comparison to XFI does not seem reasonable at this point in the manuscript. In contrast to XFI, optical based imaging methods, like BLI, suffer strongly from depth limitations.

„While PET and BLI have been already applied in MNP biodistribution studies, there are also new approaches for in vivo tracking studies like magnetic particle imaging (MPI) as in vivo imaging modality [73]. Exemplary, Wijesinghe et al. introduced MM-ODT (magneto-motive optical doppler tomography) using SPIO (supramagnetic iron-oxide) nanoparticles as tracer for tracking melanoma-cells [74]. The use for MNP tracking could be possible and should be further investigated. "

4. The novelty of the method is not properly elaborated.

In our literature research, no other work investigating the biodistribution of MNPs in vivo with XFI could be found. In comparison to other in vivo imaging modalities (PET, BLI), the advantage of XFI is the possibility of investigating the labelled MNPs in long-term studies with a chronic MNP exposure in low concentrations due to a high sensitivity and negligible limitation in terms of penetration depth due to the use of hard X-rays and without the decay of the signal like in PET due to using non-radioactive labels. We have added a conclusion to highlight the novelty of the method, and we also provided a supplementary comparison (Table 5) of the three biodistribution imaging modalities in the discussion.

“5. Conclusion

XFI has the potential to become a novel method for preclinical MNP biodistribution measurements, allowing long-term studies in small animals with a chronic MNP exposure at low concentrations. Information on uptake and distribution with a low-dose exposure would contribute to the human health risk assessment when orientating on environmentally realistic MNP concentrations in experiments. For the introduced XFI tracing agent palladium, the numerical studies have shown promising low thresholds down to a few ng/mm2 in voxel mouse simulations. Future experimental XFI studies will show whether Pd-labelled MNPs can be detected in vivo with the high sensitivity predicted by the simulations. In vivo measurements should also further investigate the potential toxicity of Pd as well as Pd leakage.”

… “An overview of the in vivo tracking imaging modalities presented can be found in Table 5.“

See Table 5 in the attachment.

6. rather than other original articles, this paper is more closer to a review paper. The simulation results can be incorporated with real data.

This work provides fundamental research for facilitating a profound design study of future experimental in vivo studies. The manuscript includes proprietary original data from numerical studies. Proving the reliability of the simulation results succeeded in two steps: Firstly, we compared the results to theoretically calculated estimations and secondly, at least for the contemplation of harvested organs, we compared the results to first experimental data (T. Staufer et al., manuscript in preparation). For integration into MNP research we conducted a literature review on the existing MNP detection methods and the requirements for MNP biodistribution studies. The focus of our manuscript lies on discussing the sensitivity limits than can be expected for XFI-scans, thus provides the first basis for planning future experimental studies.

7. How successful this method is going to be when it is applied to the real scenario? Can author discuss about the real-feasibility as it is hard to understand with simulation data.

The real feasibility of the introduced method will be proven in experimental studies, nevertheless, the numerical studies conducted in our group before convinced with a high consistency in relation to the experimental results, see e.g. our work on tracking labelled immune cells in living mice (Staufer et al., Sci Rep 2023). In this work, having compared the significance of the Pd fluorescence signal in harvested organs, the simulation results showed very good agreement with experimental data. The selected parameters used in the Geant4-setup are applicable for in vivo mouse experiments at PETRA III (DESY, Hamburg, Germany). In other words, with this work and our already published work we are convinced that the predictions about MNP-tracking are as realistic as our earlier predictions using the same physics models and simulation technique. The scope of this manuscript is to present all necessary considerations before conducting experimental studies, which, of course, require animal testing proposals for which such predictions are very helpful.

Reviewer 3 Report

Comments and Suggestions for Authors

Micro- and nano-plastics (MNPs) pollution has become a pressing global environmental issue, with growing concerns regarding its impact on human health. Humans are inevitably and continuously exposed to MNPs, raising concern about their potential risk to human health. In this paper, the authors proposed X-ray fluorescence imaging (XFI) as an imaging modality to elucidate the effective bioavailability of MNPs.

Comments

1.Enhancing the literature review section is highly recommended, particularly by presenting information in tabular form rather than through extensive discussion.

2.Voxel resolution may also be used to evaluate the proposed method.

3.Include a conclusion section that may cover contributions, analysis, and limitations, and suggest future research directions.

Comments on the Quality of English Language

Minor editing of English language required

Author Response

Reaction to Review 3 for the manuscript: 

Entitled: "Numerical Study towards in vivo Tracking of Micro- /Nanoplastic based on X-ray Fluorescence Imaging"

for Biomedicines.

With ID: biomedicines-3072991

Dear Editor, dear reviewers,

We are really thankful for your very valuable feedback which we have taken step by step for further improving our manuscript. Please find below our response to all of your points (copied from the reports and given in italics). Changes in the manuscript are highlighted in blue.

Reaction to Comments 1-3

1. Enhancing the literature review section is highly recommended, particularly by presenting information in tabular form rather than through extensive discussion.

We enhanced our discussion by offering the supplementary Table 5 with a comparison of the three imaging modalities XFI, PET, and BLI which are introduced for MNP biodistribution studies.

… “An overview of the in vivo tracking imaging modalities presented can be found in Table 5.“

Table 5. With the focus on MNP biodistribution research, X-ray Fluorescence Imaging (XFI) is compared to the two other introduced imaging modalities Positron Emission Tomography (PET) and Bioluminescence Imaging (BLI).

See Table 5 in the attachment.

2. Voxel resolution may also be used to evaluate the proposed method.

The simulation`s spatial resolution is limited by the selection of the beam diameter of 1 mm. The voxel resolution (voxel half size 0.5 mm) is higher than the resolution defined by the beam. We have added this information in the section “2.2 Voxel Based Mouse Model”.

“It was implemented in Geant4 with a voxel half size of 0.5 mm and a total of 21 x 100 x 38 voxels in the simulations, as shown in Figure 1. The voxel resolution is higher than the resolution set by the beam diameter of 1 mm, and therefore does not limit the spatial resolution in the simulations.”

3. Include a conclusion section that may cover contributions, analysis, and limitations, and suggest future research directions.

We have added the section “5. Conclusion” in which we summarize the potential of XFI for MNP biodistribution research and also emphasize the importance of further experimental studies.

“5. Conclusion

XFI has the potential to become a novel method for preclinical MNP biodistribution measurements, allowing quantitative, non-invasive, and spatially high-resolution long-term studies in small animals with a chronic MNP exposure at low concentrations. Information on uptake and distribution with a low-dose exposure would contribute to the human health risk assessment when orientating on environmentally realistic MNP concentrations in experiments. For the introduced XFI tracing agent palladium, the numerical studies have shown promising low thresholds down to a few ng/mm2 in voxel mouse simulations. Future experimental XFI studies will show whether Pd-labelled MNPs can be detected in vivo with the high sensitivity predicted by the simulations. In vivo measurements should also further investigate the potential toxicity of Pd as well as Pd leakage.”

Round 2

Reviewer 1 Report

Comments and Suggestions for Authors

Review for the manuscript:

Entitled: " Numerical Study towards in vivo Tracking of Micro- /Nanoplastic based on X-ray Fluorescence Imaging"

for Biomedicines.

With ID: biomedicines-3072991.R1

General comments

Comments for the Authors,

Authors responded to my previous remarks thus the manuscript can be published.

Best regards

Reviewer 2 Report

Comments and Suggestions for Authors

Thanks for the responses.

Comments on the Quality of English Language

Moderate checking of English is required.